# Factors Affecting the Use of Smart Mobile Examination Platforms by Universities' Postgraduate Students during the COVID-19 Pandemic: An Empirical Study

Muhammad Turki Alshurideh [1,2], Barween Al Kurdi [3], Ahmad Qasim AlHamad [4], Said A. Salloum [5,6,*], Shireen Alkurdi [7], Ahlam Dehghan [1], Mohammad Abuhashesh [8] and Ra'ed Masa'deh [9]

1. Department of Management, University of Sharjah, Sharjah 27272, United Arab Emirates; malshurideh@sharjah.ac.ae (M.T.A.); ahlam.dehghan@yahoo.com (A.D.)
2. Department of Marketing, School of Business, University of Jordan, Amman 11942, Jordan
3. Department of Business Administration, Faculty of Economics and Administrative Sciences, The Hashemite University, Zarqa 13115, Jordan; barween@hu.edu.jo
4. Information Systems Department, University of Sharjah, Sharjah 27272, United Arab Emirates; aalhamad@sharjah.ac.ae
5. Machine Learning and NLP Research Group, University of Sharjah, Sharjah 27272, United Arab Emirates
6. School of Science, Engineering, and Environment, University of Salford, Manchester M5 4WT, UK
7. Humanities Department, Al-Balqa Applied University, Amman 11134, Jordan; shireenkurdi@bau.edu.jo
8. E-Marketing and Social Media Department, Princess Sumaya University for Technology (PSUT), Amman 11941, Jordan; m.abuhashesh@psut.edu.jo
9. Management Information Systems Department, School of Business, University of Jordan, Amman 11942, Jordan; r.masadeh@ju.edu.jo
* Correspondence: salloum78@live.com; Tel.: +971-507679647

**Abstract:** Recent years have seen an increasingly widespread use of online learning technologies. This has prompted universities to make huge investments in technology to augment their position in the face of extensive competition and to enhance their students' learning experience and efficiency. Numerous studies have been carried out regarding the use of online and mobile phone learning platforms. However, there are very few studies focusing on how university students will accept and adopt smartphones as a new platform for taking examinations. Many reasons, but most recently and importantly the COVID-19 pandemic, have prompted educational institutions to move toward using both online and mobile learning techniques. This study is a pioneer in examining the intention to use mobile exam platforms and the prerequisites of such intention. The purpose of this study is to expand the Technology Acceptance Model (TAM) by including four additional constructs: namely, content quality, service quality, information quality, and system quality. A self-survey method was prepared and carried out to obtain the necessary basic data. In total, 566 students from universities in the United Arab Emirates took part in this survey. Smart PLS was used to test the study constructs and the structural model. Results showed that all study hypotheses are supported and confirmed the effect of the TAM extension factors within the UAE higher education setting. These outcomes suggest that the policymakers and education developers should consider mobile exam platforms as a new assessment platform and a possible technological solution, especially when considering the distance learning concept. It is good to bear in mind that this study is initial and designed to explore using smartphones as a new platform for student examinations. Furthermore, mixed-method research is needed to check the effectiveness and the suitability of using the examination platforms, especially for postgraduate higher educational levels.

**Keywords:** mobile examination platform; United Arab Emirates; technology acceptance model; system quality; information quality; content quality; service quality

## 1. Introduction

In modern times, mobile phones are important in all aspects of life. One paper [1] reported that, globally, there are 3.39 billion internet users, with 5.11 billion mobile users. The use of smartphone technologies has been researched in different educational aspects, such as preparation for examination [2] and enhancing students' vocabulary development [3]. Other scholars have investigated the use of mobile technologies and applications in student learning [4–6], mobile blended learning [7], enhancing learner participation and transforming pedagogy [8], to conduct student voting and the enhancement of engagement and participation [9]. Some scholars have looked at smartphone applications in the medical field [10,11] and those of engineering and technical education [12]. There is little investigation, however, into the use of the mobile phone as an accepted examination platform. Therefore, this study aims to shed some light on this subject. While studying mobile examination platforms is essential for all education stakeholders, it is particularly important to students and faculty members. Mobile examination platforms provide candidates with the means to take their exams on their phones at a time that is most suitable for them [2]. Conveniently, examiners can start the exam using mobile technology from any location [13]. These platforms are gaining in popularity because it is very easy to access them at any time [14]. In addition to examination platforms, mobile applications can be used for multiple other purposes, such as paper assessment, knowledge sharing, voting, and student registration [15,16].

### 1.1. Actual Use of Mobile Examination Platforms

As mentioned by one paper [17], information technology and innovation have become an undeniable and important part of the educational process [18]. Al-Hakeem et al. [19] stated that the online examination system is suitable for distance learning since containing the virtual appearance of lecturers and students appropriately. Within the same theme, students can use mobile apps to take an exam from distant locations [20]. Additionally, mobile examination platforms are used to monitor the academic progress of students [21]. As opined by Sung et al. [22], mobile exam platforms assist teachers to evaluate the theoretical and practical knowledge of students without concerning themselves with time and venue. Although desktop and tablet computers offer high bandwidth display and far better interactivity than smartphones, Lim [23] supported this concept because it eases the process of web-based learning by discarding the usage of desktop and tablets. The integration of mobile examination platforms has positively affected the academic performance of the student. As stated by Nikou et al. [24], portability, wireless communication, and sensitivity give the platforms the advantage over the traditional classroom examination system. For instance, the mobile examination platform "Kahoot" is used for educational purposes, serving to conduct live quizzes in class to assess student learning. The platform helps to prepare questions and distribute them among the students to assess the growth of their learning skills. Kahoot provides a range of question approaches, such as polls, quizzes, puzzles, and slides. It also makes the evaluation process easier for both teachers and students. Teachers support the usage of the mobile examination platform because it automatically calculates the grades of students and it publishes the results after the exam without consuming time [25]. According to Nikou et al. [24], students can also ascertain their practice level and take necessary steps to enhance their academic performance. Lalitha et al. [26] observed that mobile exam platforms offer user acceptance services and reduce the chances of copy-pasting and cheating to a large extent. Kaiiali et al. [27] added that mobile examination platforms control user privacy and prevent the opening of any other window until the exam is completed.

### 1.2. The Importance of Mobile Examination Platforms

Shyshkanova et al. [28] listed some of the advantageous features of mobile examination platforms, saying that they save both time and money, as well as offering security, confidentiality, and accessibility. Katz [29] pointed out that, with the help of the mobile

examination platform, instructors are relieved of the task of creating exam papers and having to arrange an examination venue and timeslots. Han et al. [30] highlighted the security and confidentiality features of the platforms. These are critical because they help in retaining the integrity of the exam and assist in evaluating the actual academic performance of students [31]. Kadam et al. [32] pointed out that any leakage from the online platform would compromise standards. However, the mobile exam platform assures the maintenance of security and confidentiality [33]. Furthermore, the existing literature shows that the mobile examination platform also offers statistical analysis of students based on their performance [34]. As stated by Chang et al. [35], the platform provides a student performance graph after the end of the exam so that both students and teachers can use it for evaluation purposes and feedback. The conduction of online exams via mobile is cheaper since there are no printing and paper costs incurred [36]. Administrators wishing to decrease expenses are likely to favor the transition from paper copy exams to the use of mobile exam platforms [24]. Another benefit of using mobile examination platforms is that they help us to save time [37]. The lengthy processes and formalities involved in formulating question papers, registering the students for exams, result declaration, and evaluation of the answer sheet are dispensed with completely with the mobile examination platforms [38].

This paper is organized as follows. We begin by introducing the literature that frames our conceptualization, followed by the development of research hypotheses. Then, we describe our research methodology and empirical results. We conclude the article by discussing the implications of the research findings for both theory and practice.

## 2. Background

### 2.1. System Characteristics

2.1.1. Quality of System

Although researchers have failed to offer a uniform definition of the quality of the system, many of them consider it to refer to system accessibility, response time, and information quality. In this context, Alshurideh et al. [39] stated that perceived usefulness, customers' acceptance, and ease of use are major criteria of quality for any Internet system. Aghazamani [40] found that features of a website or Internet system are the primary aspects affecting its level of acceptance. In this case, TAM's main elements were found to significantly mediate the behavior of Internet system users [41]. Additionally, system quality factors can be seen as the most essential elements affecting internet-based services, such as mobile cloud services, mobile exams, mobile commerce, and mobile learning [42]. Alshurideh et al. [39] opined that the quality of the system significantly affects information quality and thereby customer satisfaction in the long run. Sife et al. [43] also found that service quality is influenced directly by the information available on the Internet and is measured mostly by the quality of the information. Based on the above explanations, the quality of the system effect on both perceived usefulness and perceived ease of use can be drawn as:

**Hypothesis 1a (H1a).** *System Quality (SYS) of mobile examination platforms has a significant positive effect on their perceived usefulness (PU).*

**Hypothesis 1b (H1b).** *System Quality (SYS) of mobile examination platforms has a significant positive effect on their perceived ease of use (PEOU).*

2.1.2. Information Quality

In all forms of business, the improvement of service quality remains a primary necessity, as it fosters both revenues and growth rates [44]. In another study, Salloum et al. [45] found that information quality is the salient factor that helps in predicting customer behavior and decision-making. Evans et al. [46] opined that information quality, perceived usefulness, and attitudes are major indicators that help in predicting the purchase behavior

of customers. Based on this, Salloum et al. [47] stated that the enhancement of service quality goes hand in hand with that of information quality. These days, most organizations use the Internet to reach a wide range of customers and to increase their engagement in low-cost advertising [48]. However, the quality of information shared via the Internet remains a major concern and dilemma [49,50]. Furthermore, Al-Qaysi et al. [51] found that in such situations, the individual's acceptance is strongly influenced by information quality and response time. Accordingly, information quality is also found to be a major intrinsic motivation for using computers and the Internet in the workplace and has remained the preliminary driver of several mobile services today [52]. Based on previous explanations, the hypotheses can be drawn as:

**Hypothesis 2a (H2a).** *Information quality (INF) of mobile examination platforms has a significant positive effect on their perceived usefulness (PU).*

**Hypothesis 2b (H2b).** *Information quality (INF) of mobile examination platforms has a significant positive effect on their perceived ease of use (PEOU).*

### 2.1.3. Content Quality

According to Bates et al. [53], improving the quality of the learning environment is imperative for enhancing e-learning efficiency. Chang et al. [54] stated that the learning environment primarily includes learning content, interaction, and learning management systems offered by different e-learning systems. Content quality is, therefore, another major aspect affecting the ease of use and perceived usefulness of different mobile and Internet applications [55]. Additionally, Chen [56] found through significant investigations that content quality also impacts information quality, affects behavioral intentions of customers, and primarily consists of three dimensions which are information content, perceived ease of use, and perceived usefulness, according to Chou et al. [57]. In the case of e-learning, course quality, information, or content quality majorly assist users to promote perceived ease of use and perceived usefulness of mobile use [58]. Pituch et al. [59] particularly specified that improving content quality is important for increasing perceived web quality and interactivity. Based on the above explanations, the content quality effect can be drawn as:

**Hypothesis 3a (H3a).** *Content quality (CONT) of mobile examination platforms has a significant positive effect on their perceived usefulness (PU).*

**Hypothesis 3b (H3b).** *Content quality (CONT) of mobile examination platforms has a significant positive effect on their perceived ease of use (PEOU).*

### 2.1.4. Service Quality

According to the Technology Acceptance Model (TAM), both perceived usefulness and perceived ease of use as primary factors required for its effective use, as well as all quality keys associated with customer-centric services [60]. Gachago et al. [61] affirmed that improving service quality remains the primary aim and objective for all businesses, as it has major implications on overall productivity and profitability. It must be noted that enhancing service quality requires attention to a set of major dimensions [62]. Some of these are accessibility, the usefulness of the content, interaction, adequacy of information, and usability [63]. These factors also play a critical role in the case of e-commerce, suggesting that improving service quality is a primary necessity for enhancing e-commerce [64–66]. According to Davis [67], system and information quality are regarded as major determinants of perceived ease of use and perceived usefulness of any data or information. Based on the above explanations, the effect of service quality on both perceived usefulness and perceived ease of use can be drawn as:

**Hypothesis 4a (H4a).** *Service quality (SERV) of mobile examination platforms has a significant positive effect on their perceived usefulness (PU).*

**Hypothesis 4b (H4b).** *Service quality (SERV) of mobile examination platforms has a significant positive effect on their perceived ease of use (PEOU).*

*2.2. The Technology Acceptance Model and User Beliefs*

2.2.1. Perceived Ease of Use (PEOU)

As stated by Prestridge [68], perceived ease of use (PEOU) may be understood as the specific degree to which people believe using a certain system will be free of any effort. This measure largely facilitates new technology adoption and affects behavioral intention of using different social networks [69]. PEOU also tends to affect perceived usefulness [70]. Here, Keller et al. [71] identified that in the case of mobile learning and online course delivery systems, mentors influence students' PEOU [72]. Additionally, Palmer [73] found that social influence affects users' PEOU, suggesting that the two share an intricate relationship. Based on the above explanation, the relationship effect of PEOU on both perceived usefulness and intention to use mobile examination platforms can be expressed through the following hypotheses:

**Hypothesis 5 (H5).** *Perceived ease of use (PEOU) has a significant positive effect on the perceived usefulness (PU).*

**Hypothesis 6 (H6).** *Perceived ease of use (PEOU) has a significant positive effect on the intention to use mobile exam platforms (INT).*

2.2.2. Perceived Usefulness (PU)

According to Lee et al. [74], perceived usefulness (PU) may be considered the specific degree to which individuals believe that adopting a certain system will enhance overall job performance. PU also affects the behavioral intention of individuals to use particular social networks and is related to PEOU [75]. In the case of e-learning and mobile learning, PU is primarily affected by the instructor and the mentor, as well as social influence [76]. In this context, Lin [77] also argued that the level of satisfaction and PU influences users' continuous intention. Here, the overall satisfaction level is dependent on consumers' confirmation of expectations [78]. Based on the above explanation, the relationship effect of perceived usefulness on the intention to use mobile examination platforms can be drawn through the hypothesis detailed below.

It is well-known that mobile devices are being increasingly used as platforms for different interactive services [79]. Mobile exams and mobile learning management systems are two major services in this context, which help students in their academic endeavors [80]. In such a context, Joo et al. [81] found that exams administered via a smartphone are less expensive than conventional exams and have less scope for error, factors which promote students' preference of mobile exams over manual testing. Moreover, mobile exams are more data-driven, quick, and efficient [82]. Han et al. [30] also found that mobile exams offer more security, provide quick results, and are compatible with different subjects and streams. Moreover, the automated tests are reusable, and therefore, allow students to strengthen their foundations by taking multiple tests [83]. Accordingly, the intention to use mobile exams is, therefore, prevalent in almost all countries today, owing to rising Internet usage and other online technologies in addition to mobile technologies [84]. Based on the above discussion, the researchers hypothesize:

**Hypothesis 7 (H7).** *Perceived usefulness (PU) has a significant positive effect on the intention to use mobile exam platforms (INT).*

Based on explaining the above main study factors and the logical relations among them, the study model is illustrated here in Figure 1.

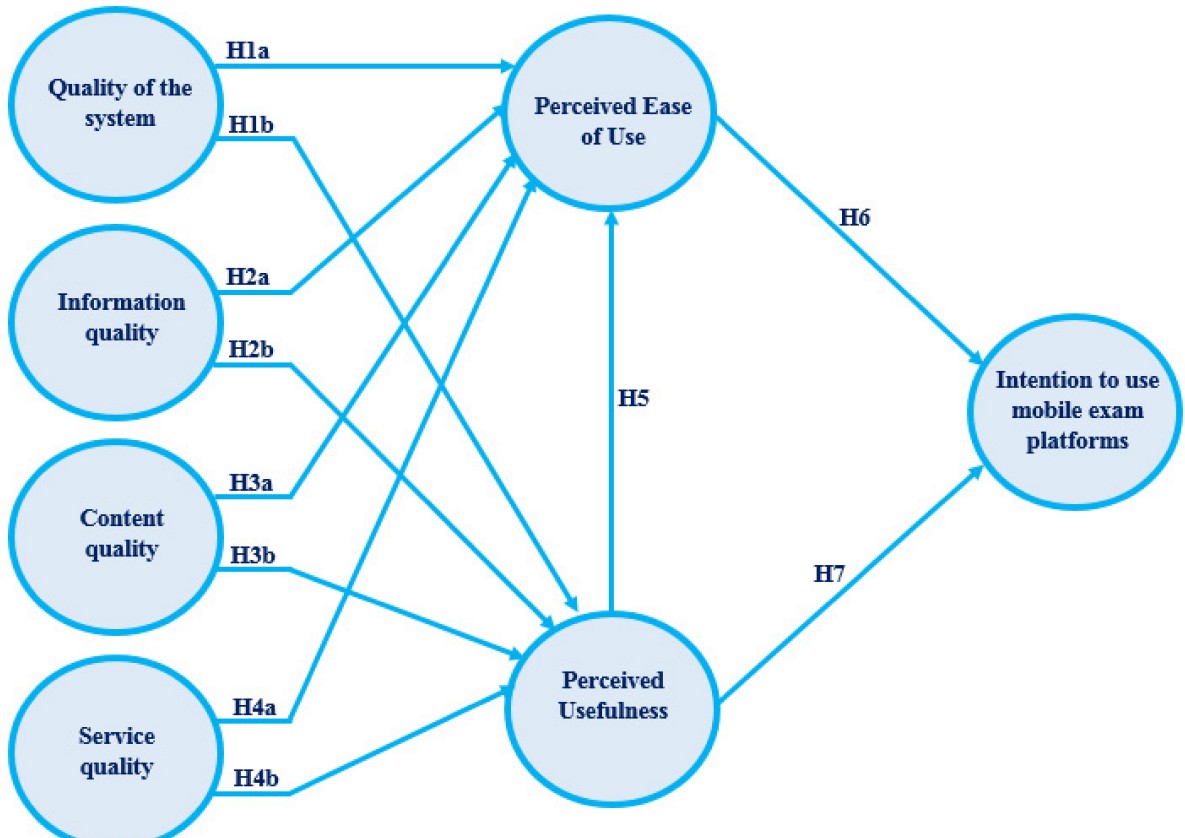

**Figure 1.** Research Model.

## 3. Materials and Methods

This section gives details regarding the data collection, the study instruments used, the survey structure, the pilot testing of the study constructs, and the study sample and its demographic data.

### 3.1. Data Collection

During the fall semester, between 15th September and 20th October 2020, the research team randomly distributed a total number of 600 hardcopy questionnaires among university students in the United Arab Emirates. Valid responses were received for 566 questionnaires, representing a total response rate of 94%. Certain missing values led to the rejection of 34 of these completed questionnaires. Hence, the team considered 566 properly filled and effective questionnaires, a figure which, according to Krejcie et al. [85], is an appropriate sample size level. Therefore, the assessment with structural equation modeling is acceptable as a sample size [86], which was subsequently employed for confirming the hypotheses. It is significant to note that present theories acted as the foundation for the hypotheses along with the incorporation of the Mobile-learning (M-learning) context. In order to assess the measurement model, structural equation modeling (SEM) (SmartPLS Version 3.2.7) was used by the group of researchers to examine the causal hypotheses based on the recommendation of [87]. For the improved action, the final path model was used.

### 3.2. Study Instrument

This research declared a survey instrument to validate the hypothesis. Intending to measure the seven constructs in the questionnaire, the survey incorporated more than

26 items. The sources of these constructs are shown in Table 1. To ensure the applicability of the study, the researchers made adjustments to questions from earlier studies.

**Table 1.** Constructs and their sources.

| Constructs | Number of Items | Source |
|:---:|:---:|:---:|
| INT | 2 | [4] |
| CONT | 4 | [88–90] |
| INF | 4 | [91–94] |
| PEOU | 4 | [88,95,96] |
| PU | 4 | [88,91,95,96] |
| SYS | 4 | [91–93,96,97] |
| SERV | 4 | [90–92,94] |

Note: INT = intention to use mobile examination platforms; CONT = content quality; INF = Information quality; PEOU = perceived ease of use; PU = perceived usefulness; SYS = quality of the system; SERV = service quality.

### 3.3. Survey Structure

The students were provided with and asked to complete a questionnaire survey. The survey was divided into three sections:

1. The first section concerned the personal data of the participants.
2. The second section focused on the five items illustrating the general question regarding mobile-learning systems.
3. The third section contained the 15 items that show Intention to use mobile examination platforms, Content quality, Information quality, Perceived Ease of Use, Perceived Usefulness, Quality of the system, and Service quality.

The 26 items were measured through a five-point Likert Scale with the following values: (1) Strongly disagree, (2) Disagree, (3) Neutral, (4) Agree, and (5) Strongly agree.

### 3.4. A Pilot Study of the Study Constructs

A pilot study helped to conclude the reliability of the questionnaire items. For the pilot study, about 60 students were selected at random from the population. The sample size comprised 600 students and this was 10% of the total sample size of this research. Additionally, the criterion was closely followed. In order to assess the outcomes of the pilot study, the Cronbach's alpha test was employed along with the help of IBM SPSS Statistics Version 23 (IBM, Armonk, NY, USA) for internal reliability. Thus, all the suitable conclusions for the measurement items were drawn. If the recommended outline of social science research studies is followed [98], then the reliability coefficient of 0.7 is deemed to be acceptable. Table 2 shows the Cronbach alpha values for the following seven measurement scales.

**Table 2.** The pilot study.

| Construct | Cronbach's Alpha |
|:---:|:---:|
| INT | 0.868 |
| CONT | 0.882 |
| INF | 0.829 |
| PEOU | 0.799 |
| PU | 0.836 |
| SYS | 0.890 |
| SERV | 0.845 |

Note: INT = intention to use mobile examination platforms; CONT = content quality; INF = Information quality; PEOU = perceived ease of use; PU = perceived usefulness; SYS = quality of the system; SERV = service quality.

### 3.5. The Study Sample

The research team circulated hard copies of the questionnaire survey to students at a number of different universities in the United Arab Emirates (UAE) (N = 600).

### 3.6. The Study Sample's Demographic Data

Table 3 encapsulates the study participants' personal/demographic data. The ratio of male to female students was 52% to 48%, respectively. A total of 57% of the respondents fell into the age category of between 18 and 29 years, while 43% of the respondents were above 29 years old. Regarding the students' academic majors, 43% studied Business Administration, 23% were enrolled in the College of Engineering and Information Technology, and 19% were enrolled in the College of Mass Communication and Public Relations, while 9% were students of General Education and 6% of Humanities and Social Sciences. All of the respondents were from an educated background and were in pursuit of a university degree. A total of 70% of the respondents held a Bachelor's degree, while 19% possessed a Master's degree. Furthermore, 11% of the respondents were holders of a doctoral degree, while the remainder were diploma holders. According to Al-Emran et al. [99], the "purposive sampling approach" is appropriate when access to the respondents is easy and they are willing to volunteer. The study sample was made up of students from different colleges, of different ages, and studying at different levels. In addition, the demographic data were measured with the help of IBM SPSS Statistics Version 23. The comprehensive demographic data of the respondents are shown in Table 3.

**Table 3.** Respondents' demographic data.

| Criterion | Factor | Frequency | Percentage |
|---|---|---|---|
| Gender | Female | 271 | 48% |
| | Male | 295 | 52% |
| Age | 18–29 | 320 | 57% |
| | 30–39 | 190 | 34% |
| | 40–49 | 48 | 8% |
| | 50–59 | 8 | 1% |
| College | College of Business Administration | 242 | 43% |
| | College of Humanities and Social Sciences | 35 | 6% |
| | College of Engineering and Information Technology | 130 | 23% |
| | College of General Education | 51 | 9% |
| | College of Mass Communication and Public Relations | 108 | 19% |
| Education qualification | Bachelor's | 395 | 70% |
| | Master's | 105 | 19% |
| | Doctorate | 66 | 11% |

## 4. Results and Discussion

### 4.1. Data Analysis

To conduct the data analysis, the partial least squares-structural equation modeling (PLS-SEM) was used with the aid of SmartPLS V.3.2.7 software in this research [100]. To analyze the collected data, a two-step assessment approach was used that consists of a structural model and measurement model [66]. For this research, PLS-SEM is considered to be most suitable [101]. PLS-SEM [87] will help to deal with the investigative studies that consist of complex models. It also analyzes the whole model in one go [102]. PLS-SEM provides the concurrent analysis for both the measurement and structural model, which will also give accurate calculations [103].

#### 4.1.1. Convergent Validity

Validity (having convergent and discriminant validity) and the construct reliability (including composite reliability (CR), Dijkstra-Henseler's rho (pA), and Cronbach's alpha

(CA)) are taken into account for the evaluation of measurement model as stated by Hair et al. [66]. Table 4 illustrates that Cronbach's alpha (CA) has the values between 0.718 and 0.897 in order to identify the construct reliability. These values surpass the threshold that is 0.7 [104]. The findings in Table 4 also shows that the values from 0.755 and 0.903 are part of the composite reliability (CR) and it is evident that these values are more than the threshold of 0.7 [105]. Thus, the researchers must use Dijkstra-Henseler's rho (pA) reliability coefficient [106] in order to assess the construct reliability. In investigative research the reliability coefficient $\varrho$A values must be more than 0.7, similar to CA and CR, while values higher than 0.8 and 0.9 are used in later stages of study [104,107,108]. The reliability coefficient $\varrho$A of each measurement construct is bigger than 0.70 according to Table 4. It was presumed that all the constructs are accurate at reaching the final stage and the construct reliability has been checked against these findings.

**Table 4.** Convergent validity results that assure acceptable values (Factor loading, Cronbach's Alpha, composite reliability, Dijkstra-Henseler's rho $\geq$ 0.70 and AVE > 0.5).

| Constructs | Items | Factor Loading | Cronbach's Alpha | CR | PA | AVE |
|---|---|---|---|---|---|---|
| Intention to use mobile exam platforms | INT1 | 0.799 | 0.815 | 0.828 | 0.821 | 0.625 |
| | INT2 | 0.728 | | | | |
| Content quality | CONT1 | 0.758 | 0.718 | 0.755 | 0.780 | 0.661 |
| | CONT2 | 0.865 | | | | |
| | CONT3 | 0.859 | | | | |
| | CONT4 | 0.796 | | | | |
| Information quality | INF1 | 0.839 | 0.753 | 0.801 | 0.798 | 0.650 |
| | INF2 | 0.887 | | | | |
| | INF3 | 0.740 | | | | |
| | INF4 | 0.822 | | | | |
| Perceived Ease of Use | PEOU1 | 0.730 | 0.869 | 0.819 | 0.836 | 0.612 |
| | PEOU2 | 0.777 | | | | |
| | PEOU3 | 0.885 | | | | |
| | PEOU4 | 0.848 | | | | |
| Perceived Usefulness | PU1 | 0.799 | 0.852 | 0.903 | 0.894 | 0.709 |
| | PU2 | 0.868 | | | | |
| | PU3 | 0.912 | | | | |
| | PU4 | 0.820 | | | | |
| Quality of the system | SYS1 | 0.760 | 0.806 | 0.887 | 0.889 | 0.598 |
| | SYS2 | 0.850 | | | | |
| | SYS3 | 0.884 | | | | |
| | SYS4 | 0.826 | | | | |
| Service quality | SERV1 | 0.731 | 0.897 | 0.839 | 0.842 | 0.741 |
| | SERV2 | 0.882 | | | | |
| | SERV3 | 0.851 | | | | |
| | SERV4 | 0.844 | | | | |

According to Hair et al. [66], in order to carry out measurement of the convergent validity, we need to assess average variance extracted (AVE) and factor loading. Table 4 suggests that the value of 0.7 is still lesser than the factor loading values. While Table 1

has shown that the values provided by AVE that are from 0.598 and 0.741 are the ones that are greater than the threshold value of '0.5,' the success in attaining convergent validity is dependent on the expected outcomes.

### 4.1.2. Discriminant Validity

In order to undertake the measurement of discriminant validity [66], it was suggested to measure two standards: the Fornell–Larker principle and the Heterotrait–Monotrait ratio (HTMT). As shown in Table 5 [109], the Fornell–Larker principle has verified the obligations as all the AVEs and their square roots are more than its correlations with other models.

**Table 5.** Fornell–Larker Scale.

|  | INT | CONT | INF | PEOU | PU | SYS | SERV |
|---|---|---|---|---|---|---|---|
| INT | 0.798 * | | | | | | |
| CONT | 0.430 | 0.852 | | | | | |
| INF | 0.518 | 0.459 | 0.817 | | | | |
| PEOU | 0.514 | 0.600 | 0.528 | 0.832 | | | |
| PU | 0.268 | 0.225 | 0.458 | 0.336 | 0.859 | | |
| SYS | 0.328 | 0.158 | 0.316 | 0.125 | 0.158 | 0.874 | |
| SERV | 0.520 | 0.105 | 0.444 | 0.540 | 0.487 | 0.230 | 0.785 |

Note: INT = intention to use mobile exam platforms; CONT = content quality; INF = Information quality; PEOU = perceived ease of use; PU = perceived usefulness; SYS = quality of the system; SERV = service quality. * Diagonals (bold values) represent the square root of average variance extracted, and the other matrix entries are the factor correlation.

In Table 6, the HTMT ratio outcomes are shown, illustrating that the threshold value of 0.85 is still above the value of every construct [69], leading to the establishment of the HTMT ratio. These findings help to know the discriminant validity. The results of the assessment show that there were no problems about the validity and reliability were faced during the measurement model's evaluation. Thus, to use the collected data more productively, the structural model can be judged.

**Table 6.** Heterotrait–Monotrait Ratio (HTMT).

|  | INT | CONT | INF | PEOU | PU | SYS | SERV |
|---|---|---|---|---|---|---|---|
| INT | | | | | | | |
| CONT | 0.200 | | | | | | |
| INF | 0.652 | 0.698 | | | | | |
| PEOU | 0.550 | 0.605 | 0.408 | | | | |
| PU | 0.391 | 0.300 | 0.399 | 0.105 | | | |
| SYS | 0.205 | 0.574 | 0.498 | 0.618 | 0.501 | | |
| SERV | 0.299 | 0.505 | 0.345 | 0.700 | 0.544 | 0.229 | |

Note: INT = intention to use mobile examination platforms; CONT = content quality; INF = Information quality; PEOU = perceived ease of use; PU = perceived usefulness; SYS = quality of the system; SERV = service quality.

### 4.2. Model Fit

The standard root mean square residual (SRMR), exact fit criteria, d_ULS, d_G, Chi-Square, NFI, and RMS_theta are the fit measures provided by the Smart PLS that demonstrate the model fit in PLS-SEM [110]. The difference between the expressed correlations and the correlations from the model that made use of the correlation matrix [87] in accordance with the SRMR, of which the values are considered as good model fit measures [111] when they are below 0.08, while NFI values higher than 0.90 are considered as the model fit [112]. The ratio of the Chi2 value of the proposed model to the null/benchmark model is the NFI [113]. The NFI is not suitable to be the model fit measure since the larger the parameters are, the higher the NFI is Hair et al. [87]. The two metrics are: squared Euclidean distance, d_ULS, and the geodesic distance d_G, which help to find any discrepancy between the empirical covariance matrix and the covariance matrix as understood by the

composite factor model [87,106]. Only in the reflective models, the RMS theta can be implied, and this will appraise the outer model residuals' correlation degree [113]. If the values are lower than 0.12, they will be known as a good fit, and when the RMS theta value is nearer to zero, the PLS-SEM model will be considered as better; otherwise, the values will not show a good fit [114]. It was recommended by Hair et al. [87] that the estimated model considers the total impact and model structures; on the other hand, the saturated model assesses the connection between all constructs.

According to Table 7, the RMS Theta value was around 0.082, which means that in order to exhibit the global PLS model validity, the required goodness-of-fit for the PLS-SEM model is sufficient.

**Table 7.** Model fit indicators.

| Criteria | Complete Model | |
| --- | --- | --- |
| | **Saturated Model** | **Estimated Mod** |
| **SRMR** | 0.042 | 0.050 |
| **d_ULS** | 0.895 | 2.408 |
| **d_G** | 0.677 | 0.626 |
| **Chi-Square** | 470.827 | 482.459 |
| **NFI** | 0.715 | 0.738 |
| **RMS Theta** | 0.082 | |

*4.3. Hypotheses Testing—Path Coefficient*

When the measurement model is confirmed, the next step is the structural model [115–120]. Through a bootstrapping procedure containing 5000 re-samples, this involves determining the path coefficients and the coefficient of determination ($R^2$) [66]. The structural equation model had a high predictive power, as shown in Figure 2 and Table 8 [121], which also shows that the variance's percentages, i.e., almost 71%, 72%, and 73%, are the percentage of the variance in the perceived usefulness, perceived ease of use, and intention to use mobile examination platforms, respectively. This model was used along with Smart PLS and had a maximum likelihood estimation in order to know the interdependence of a range of theoretical constructs of the structural model [87,122,123]. Concerning the path analysis, the path coefficients, *t*-values, and *p*-values for each hypothesis are shown in Table 9, and all hypotheses have been supported. Based on the data analysis hypotheses H1a, H1b, H2a, H2b, H3a, H3b, H4a, H4b, H5, H6, and H7 were supported by the empirical data. The Quality of the system (SYS), Information quality (INF), Content quality (CONT), and Service quality (SERV) have significant effects on the Perceived Ease of Use (PEOU): β = 0.436, $p < 0.001$, β = 0.769, $p < 0.001$, β = 0.158, $p < 0.05$, β = 0.318, $p < 0.05$, respectively; hence, H1a, H2a, H3a, and H4a are supported. The Quality of the system (SYS), Information quality (INF), Content quality (CONT), and Service quality (SERV) also have significant effects on the Perceived Usefulness (PU): β = 0.287, $p < 0.001$, β = 0.335, $p < 0.001$, β = 0.789, $p < 0.001$, β = 0.531, $p < 0.001$, respectively; hence, H1b, H2b, H3b, and H4b are supported. Finally, the results also showed that the Perceived Ease of Use (PEOU) significantly influenced the Perceived Usefulness (PU) (β = 0.262, $p < 0.001$) and Intention to use mobile exam platforms (INT) (β = 0.487, $p < 0.001$), supporting hypotheses H5 and H6, respectively. The Perceived Usefulness (PU) was determined to be significant in affecting the Intention to use mobile examination platforms (INT) (β = 0.366, $p < 0.001$), supporting hypothesis H7.

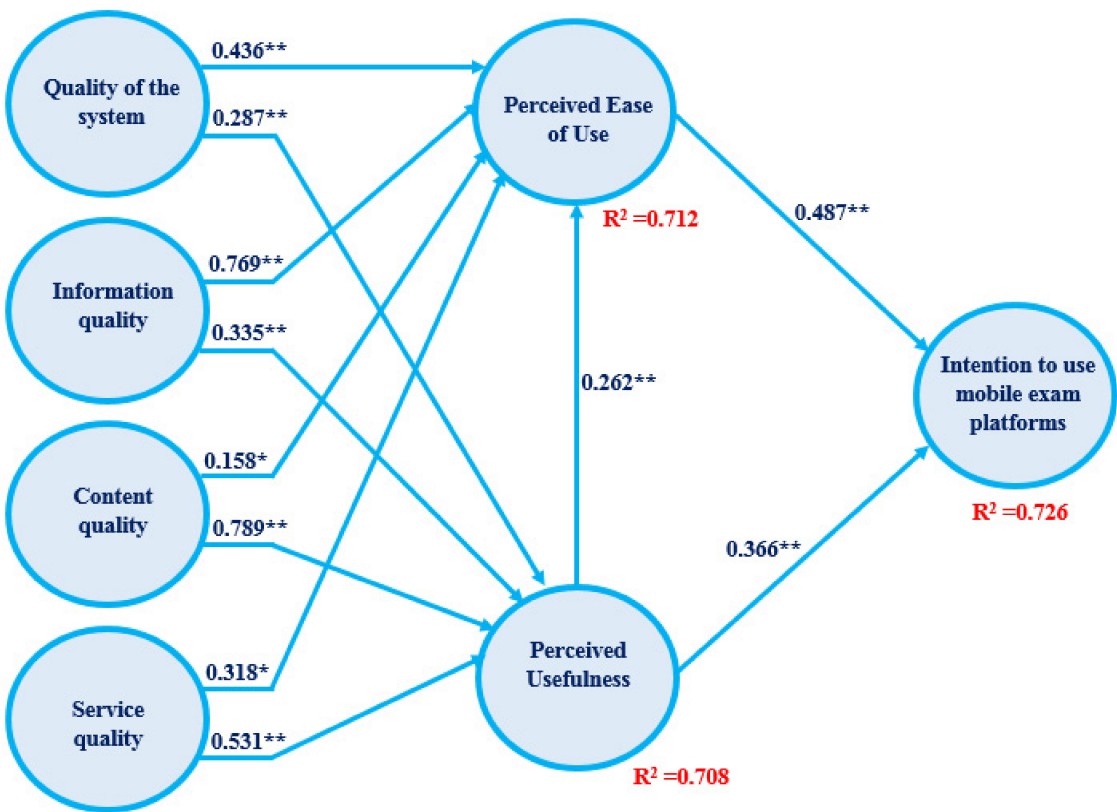

**Figure 2.** Hypotheses' testing results. * $p < 0.05$, ** $p < 0.01$.

**Table 8.** $R^2$ of the endogenous latent variables.

| Constructs | $R^2$ | Results |
|:---:|:---:|:---:|
| INT | 0.726 | High |
| PEOU | 0.719 | High |
| PU | 0.708 | High |

**Note:** INT = intention to use mobile examination platforms; PEOU = perceived ease of use; PU = perceived usefulness.

**Table 9.** Results of structural model examination (significant at * $p < 0.05$, ** $p < 0.01$).

| Hypothesis | Relationship | Path | T-Value | *Path Coefficient* | Result |
|:---:|:---:|:---:|:---:|:---:|:---:|
| H1a | SYS -> PEOU | 0.436 | 24.635 | +0.000 | Accepted ** |
| H1b | SYS -> PU | 0.287 | 18.009 | +0.000 | Accepted ** |
| H2a | INF -> PEOU | 0.769 | 15.546 | +0.000 | Accepted ** |
| H2b | INF -> PU | 0.335 | 10.222 | +0.000 | Accepted ** |
| H3a | CONT -> PEOU | 0.158 | 2.521 | +0.022 | Accepted * |
| H3b | CONT -> PU | 0.789 | 9.445 | +0.003 | Accepted ** |
| H4a | SERV -> PEOU | 0.318 | 1.630 | +0.026 | Accepted * |
| H4b | SERV -> PU | 0.531 | 13.780 | +0.000 | Accepted ** |
| H5 | PEOU -> PU | 0.262 | 11.248 | +0.000 | Accepted ** |
| H6 | PEOU -> INT | 0.487 | 13.990 | +0.000 | Accepted ** |
| H7 | PU -> INT | 0.366 | 10.201 | +0.001 | Accepted ** |

Note: INT = intention to use mobile examination platforms; CONT = content quality; INF = Information quality; PEOU = perceived ease of use; PU = perceived usefulness; SYS = quality of the system; SERV = service quality.

## 5. Conclusions

The data gathered clearly indicate that the majority of the study sample considered mobile learning platforms to be a convenient tool of assessment. Among the study sample, participants of the age group 18–29 years particularly expressed interest in using mobile

examination platforms, which could help in adopting new assessment techniques, which in turn makes the assessment process easier.

The results show that the main parameter that promotes students' use of mobile examination platforms is system quality. If users find the quality of the system to be high, their willingness and intention to use such new examining approaches properly are boosted. This confirms the view of Akar et al. [42], who saw system quality as the most essential element affecting both Internet- and mobile-based services, such as mobile cloud services, mobile learning and exams services, and even mobile commerce services. Moreover, this study found that information quality plays an essential role in both perceived ease of use and perceived usefulness of mobile examination platforms. Many scholars confirmed these results. Davis [67], for example, declared that information quality is regarded as one of the major determinants of the perceived ease of use and perceived usefulness of any data or information used. Moreover, the collected data and results show that the quality of the content of both mobile learning and mobile exam platforms also affect their usefulness. The comprehensive and superior quality content of mobile learning and mobile examination platforms helps students in acquiring subtle knowledge and test such knowledge directly in any taught subject. The comprehensiveness and superior quality of the content help students to master the subject matter, especially seeing as access is flexible and they can read the topics at a convenient time, and examine themselves many times accordingly. Thus, for potential users, the better use of mobile examination platforms comes by improving the quality of the content, which in turn helps to maximize the users' benefits and practices.

Service quality was found to influence both ease of use and perceived usefulness of mobile examination platforms. According to Freeze et al. [60], both perceived ease of use and perceived usefulness are primary indicators for the effective use for any system and the quality of such a system was found essential for customer-centric provided services. Additionally, according to Simonova [12], Al-Dweeri et al. [124] and Al Dmour et al. [125], improving the service quality remains the primary aim for business organizations that provide a wide range of services, especially those using mobile service applications. Accordingly, it becomes clear that mobile examination platforms have made the learning process convenient for the majority of students in different disciplines, such as engineering, medicine, business, and Information Technology. While these platforms can be used for taking online exams, they also serve to enhance the innovative learning platforms through hosting brainstorming sessions and holding interactive lectures. A good example of that is mentioned by Akour et al. [126] and Bacca-Acosta et al. [127], which in turn helps in enhancing students' retention and satisfaction [128–130]. System quality and content quality are found to be prerequisite drivers that affect students' acceptance and adoption of mobile examination and learning applications, as declared by Liu et al. [131]. The offered system quality and the quality of the content help students to better perceive the level to which a particular mobile examination application can be useful to them and how user-friendly it is. This issue is discussed and confirmed by many scholars, such as Day et al. [132], who confirmed the need for high-quality, safe content in teaching mobile applications, especially the technical ones. Moreover, Gorla et al. [133] pointed out the necessity of high-quality IT management systems, information, and services, which in turn, affect users' ability to use mobile examination platforms efficiently.

### 5.1. Theoretical and Practical Implications of the Study and Recommendations

Manner et al. [134] looked at the theoretical implications of mobile exam platforms from the perspective of three academic disciplines, namely sociology, technology, and pedagogy. The mobile exam as a means of supporting social inclusion needs outspoken principles on what is being learned as well as what counts as the effective outcomes [135]. It is also where the constructivist education theory comes in Nikou et al. [136]. The technological needs must be developed depending on a tested and educated understanding of the technical support of mobile examination platforms [137]. The practical implications of mobile examination platforms dictate the provision of perfect and safe testing grounds

for different types of candidates [138]. Thus, additional theoretical research and more practical tests are needed to check the practicality and evaluate the performance and consequences of the applications. Mobile exam software must work dynamically to be user-friendly and to provide direct feedback to all candidates taking the test [139]. Educational institutions that wish to pursue the use of mobile phone examination platforms should invest greater amounts into developing system and service quality, and work intensively on enhancing information quality and the quality of exam content. Currently, thousands of educational institutions around the world are facing the COVID-19 pandemic and are under pressure from governments to commit to both online and blended learning. Based on this, it is evident that the hundreds of millions of students who can neither attend classes nor take part in traditional examinations would find the use of both electronic exams and mobile phone exam techniques an appropriate solution.

The introduction and increased use of mobile examination platforms by educational institutions serves to facilitate the teaching and examination processes. As opined by Al Masri [140], students can take the exams via their mobile phones at a time convenient to them, but nobody can check the exam process and evaluate the performance. The students might get the answers from the Internet, which may have an adverse effect on students' true knowledge levels [36]. Therefore, the teachers must set questions for which the answers cannot be easily accessed on the Internet or in books [141]. Furthermore, it is crucial to be able to set a timer for each question so that students do not have sufficient time to search the Internet for the answer [142]. Mobile examination platforms are susceptible to fraud [14]. Technical errors may cause some difficulties in using mobile phone examination platforms. For example, a student who encounters some system failure or smartphone malfunction may miss sitting the exam [143], or there may be some difficulties in controlling and securing the exam environment.

In times of emergency and natural crisis, all governmental institutions find themselves under pressure to carry out their functions in the best way possible under the new imposed circumstances. At this time, educational institutions are being asked to take definite steps towards planning and applying mobile phone learning and examinations technologies [144]. At the onset of the current crisis in early 2020, the use and intention to use such technologies was still in the introductory stages, and additional investment is needed to enhance the mobile phone education and examination environment and culture [145]. This study seeks to provide both theoretical and empirical approaches to understanding the drivers behind the use of the main mobile phone exam platforms and highlighting which of these drivers need to be planned for and employed properly. Earlier studies confirmed the necessity to use such mobile phone examination systems and applications [39,49] and, these days, that necessity is greater than at any other time.

*5.2. Research Limitations*

This study was conducted to investigate the main factors affecting the intention to use mobile examination platforms by university students. This study is initial and can be classified as an exploratory study to check the suitability of using smartphones as a proper platform for conducting some students' examinations. Employing smartphones as an examination platform is important to be validated and checked using mixed-method research approaches. Additionally, it is good to remember that using smartphones as an examination platform will not fit all examination levels (e.g., evaluation and criticizing) and might not be a good substitute for classical examination methods; however, they are worthy to be used and it is important to shine more light on how they can be used within academia. Accordingly, using smartphone examination platforms for testing higher education learning approaches, such as criticizing, evaluation, and even explanation, may be limited and seen as not appropriate from the instructors' point of view. Lecturers and instructors use different examination methods to check their students' understanding and knowledge. However, using such an approach for postgraduate students' examinations needs to be checked in more detail, and its effectiveness should be tested with respect to

different disciplines. A limited amount of primary data were collected for analysis. A large sample size is essential, especially to test the intention to use such platforms within different pedagogical settings. Future scholarly works regarding the use of mobile phone exam platforms could encompass a larger number of students over various levels of studies and disciplines, bearing in mind that openness to the use of mobile phone examination platforms can differ from one discipline to another. Thus, additional theoretical studies and real classroom applications are needed, especially to test the adoption of such platforms and their interrelated elements, which are system quality, information quality, and content quality. Additional factors that might be worth investigation are enjoyment and entertainment value and how such factors could potentially increase the intention, use, and repeat use of such new exam techniques. There remain obstacles to the comprehensive use of such platforms by a majority of students, considering that some students do not own a smartphone, and others may find it difficult to use the applications without help from others. However, the amount of research carried out on students' orientation towards and their experiences with using such platforms is limited, and is a potential aspect to be addressed in later studies. Moreover, it is good to apply such research on a real examination setting such as quizzes, which rely more on using some simple examination methods such as true/false or multiple-choice questions. The next step is to analyze the practical results, instead of relying on the respondents' feelings and thoughts using a Likert Scale. In other words, it is important to have practical results: this need to be investigated by using other methods, such as taking users' and instructors' view qualitatively and explore the findings using content analytical techniques to strengthen the use of a smartphone examination platform. To sum up, this study pioneers this issue, and more investigation is needed.

**Author Contributions:** Conceptualization, M.T.A. and S.A.S.; methodology, S.A.S.; software, B.A.K.; validation, A.Q.A., S.A., and A.D.; formal analysis, M.A.; investigation, R.M.; resources, M.T.A.; data curation, B.A.K.; writing—original draft preparation, S.A.; writing—review and editing, R.M.; visualization, S.A.S.; supervision, A.D.; project administration, A.Q.A.; funding acquisition, S.A. All authors have read and agreed to the published version of the manuscript.

**Funding:** This research received no external funding.

**Institutional Review Board Statement:** Not applicable.

**Informed Consent Statement:** Not applicable.

**Data Availability Statement:** The data presented in this study are available on request from the corresponding author.

**Acknowledgments:** We would like to express my appreciation to my co-authors for accepting to participate in this study, especially while they are working for five different universities, namely, the University of Sharjah, the University of Jordan, the Hashemite University, the Al-Balqa' Applied University, and the Princess Sumaya University for Technology, which are located in The United Arab Emirates and Jordan. Without them, this paper would not have been published. I would like to thank the reviewers for their great comments and valuable feedback.

**Conflicts of Interest:** The authors declare no conflict of interest.

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
