# Peer review of "Factors Affecting the Use of Smart Mobile Examination Platforms by Universities’ Postgraduate Students during the COVID-19 Pandemic: An Empirical Study"

_informatics, doi:10.3390/informatics8020032_

Round 1
Reviewer 1 Report
In general, this is an interesting work of great interest in the field of educational technology. There are certainly not too many scientific studies that focus on considering the position of students on the use of cell phones as a platform for taking exams, a topic that has been the subject of much debate at the present time due to the COVID-19 pandemic.
The study is generally rigorous and correctly meets all the characteristics of a scholarly article. There are not too many recommendations for changes that we consider pertinent to make, but we believe that the following would improve the good work already done:
1. In the abstract and on page 17 we recommend that "Coronavirus pandemic" be replaced by "COVID-19 pandemic". This is a more accurate term since it defines the disease, which is the basis of a pandemic, and not the virus.
2. In section 2, entitled "Literature review", we would advise titling it "Background", which is more appropriate for papers of this nature.
3. There is one section, section 3, "The study model", which is only based on an outline and therefore we do not recommend that it be presented as a stand-alone section. We would ask the authors to consider introducing it in what is now section 4, "Research Methodology", modifying its name and titling it "Materials and methods". Thus, after section 2 would come section 3, which would have these sections:
(New structure merging the current sections 3 and 4) 3. Materials and methods", 3.1. "The study model", 3.2. Data collection, 3.3 Study Instrument.... etc.
4. The current section 5 (which would be section 4 in the structure we propose) should be entitled, instead of "Findings and Discussion", "Results and Discussion". And, of course, section 6 (which should be section 5) should just be "Conclusions" instead of "Discussion and conclusion". There can't be two sections collecting the discussion. However, the conclusion section would be too long, so we would really advise the authors, if they can manage it, to make a separate discussion section. The traditional structure of a paper of these characteristics is: Results - Discussion - Conclusions.
5. In what there are multiple defects, and it is the main drawback of this study, is in the formal aspect regarding the work of sources, citations and references. In general, the rules of the journal are not correctly applied. Thus, for example, where it says "[35], [45]" should be "[35,45]"; where it says "[123]-[125]" should be [123-125]; where it says "[10], [119], [120]" should be "[10,119,120]"; and so on throughout the manuscript, which should be thoroughly revised.
6. The same applies to bibliographic references, which should follow the standards of the journal and MDPI. None of them use them, so they should be completely modified.
7. The titles of the sections and sections should also be harmonized, in some cases it is used to start with capital letters in all the words (as in Convergent Validity or The Study Sample) and in others only with capital letters the first one (as in Research limitations or Data analysis). It is necessary to follow the same criteria.
We are grateful for the opportunity to have reviewed this manuscript, which we hope will be an important contribution to the current knowledge of the TAM.
Author Response
The authors are really very grateful to the feedback and comments raised by the reviewer which really assist them to significantly enhance this work and its presentation. The productive and valuable remarks enable us to update many parts of the paper as shown by the responses to each comment. Our responses are mentioned below under each comment raised by the reviewer and it is written in (Times New Roman, red color). Besides, all the updated parts in the manuscript were highlighted in yellow color in order to be easily tracked by the reviewers.

Reviewer 2 Report
The manuscript "Factors Affecting the Use of Smart Mobile Exam Platforms by Universities' postgraduate students During the Covid 19 Pandemic: An Empirical Study" presents an interesting study on the use of smart mobile for exams by postgraduate students during the pandemic. The theoretical contributions are sufficient and up to date. However, there are some aspects that need to be improved:
- The methodology section needs to be clearer, for which I recommend that the authors structure it in the following sections: Participants (in this section they should include the way the sample was chosen and, if applicable, the randomization procedure used on the population sample), the final sample data should be presented in a table that includes its disaggregation by gender and the mean and standard deviation statistics, as well as the partial and total number of participants.) Instruments (this section should include all the instruments used with reliability and validity indicators). Procedure (this section should clearly and thoroughly indicate how the study was developed so that it can, if necessary, be replicated by the scientific community). Data analysis (this section should include the measurement tests used to contrast each of the hypotheses presented). Design (this section should include the type of design applied (pre-experimental, cuasi-experimental, etc.). All this information is reflected by the authors in the manuscript, but it is not adequately organized.
- Neither is the approval by a Bioethics Committee or Commission, nor the written informed consent of the participants in the study clearly explained.
- There is also confusion regarding the hypotheses, since at the beginning there are two hypotheses:
"H2a: Information quality (INF) of mobile exam platforms has a significant positive effect on their perceived usefulness (PU).
H2b: Information quality (INF) of mobile exam platforms has a significant positive effect on their perceived ease of use (PEOU)."
Then reference is made to:
"H5: Perceived ease of use (PEOU) has a positive effect on the perceived usefulness (PU) and intention to use mobile exam platforms.
H6: Perceived ease of use (PEOU) directly and significantly influences the intention to use mobile exam platforms (INT).
H7: Perceived usefulness (PU) directly and significantly influences the intention to use mobile exam platforms".
And further on it states H1a, H1b, H2a, H2b, H3a, H3b, H4a, H4b, H5, H6, and H7 p.13.
Therefore, I recommend that, once the Introduction section has been completed, the authors clearly state the hypotheses or research questions of the study and then, in the results section, present the results in an orderly fashion according to each hypothesis.
- Regarding the Good fitness used, it is recommended that the authors include all these parameters: Chi-square, RAMSEA, RAMSEA Interval, SRMR, TLI, CFI, AIC, ECVI, ECVI Interval and also that they include in the table another column in which they refer to the value accepted as correct for each of the parameters.
- As minor issues, it is recommended that the authors review the following aspects:
- Presentation of the tables according to the journal's guidelines.
- Improve the elaboration and resolution of the figures according to the journal's standards.
- Review all the bibliographic references to check that they all have year and concept (see e.g. reference 107, 138) and present them according to the journal's standards.
Author Response

(The authors gave the same response as above.)

Reviewer 3 Report
Abstract
The sentence "Results showed that all study hypotheses are supported and confirmed the effect of the TAM extension factors" seems to imply that TAM is a causal factor rather than a passive conceptual model.
The concluding suggestion, that "policymakers and education developers should consider mobile exam platforms as a new learning platform" is puzzling and is certainly not supported by the research, which focused on assessment, not learning.
1. Introduction
The banal sentence "Without a doubt, we can affirm that we are now living in the mobile era." contributes nothing and is out of place in an academic journal.
The informal use of "exam" (p. 1) should be corrected to "examination" throughout the paper, including the title.
In §1.1 the statement "the online examination system is good for distance learning" lacks academic critique and should be qualified: "good" in what ways?
In §1.1 it is claimed that the use of smartphones "eases the process of web-based learning by discarding the usage of desktop and tablets". However, desktop and tablet computers offer high bandwidth display and far better interactivity than smartphones, which are limited to a smaller sized interface and text-based assessment such as quizzes. This restricts their use to Remember and Understand levels of learning (see https://tips.uark.edu/using-blooms-taxonomy/). Apps such as Kahoot provide an impoverished educational experience that denies higher activities such as collaborative learning and applying knowledge in real-world contexts. Smartphones are definitely not a 'silver bullet', and their limitations should be acknowledged.
In §1.2, statements such as " instructors are relieved of the task of creating exam papers" and "assist in evaluating the actual academic performance of students" are uncritical and facile. They are out of place in an academic journal.
2. Literature review
In §2.1.1 the reasons for formulating H1a & H1b need to be better justified. It is not enough to selectively cite a few sources.
§2.1.2 and the H2 hypotheses concern information quality, but it is unclear how this is related to the use of smartphone quizzes to assess low-level knowledge. Surely, the 'quality' of the information presentation is already evident prior to its summative assessment; so how can that 'quality' be subsequently enhanced by the assessment format?
Similarly, for §2.1.3 and the H3 hypotheses, the 'quality' of the curriculum content has already been presented prior to its summative assessment; so how can that 'quality' be subsequently enhanced by the assessment format?
Similarly, for §2.1.4 and the H4 hypotheses, the 'quality' of the educational service is already evident prior to its summative assessment; so how can that 'quality' be subsequently enhanced by the assessment format?
3. The study model
Figure 1 is simply a theoretical factors diagram. There is no discussion in this section of the educational context for the study model.
4. Research methodology
In §4.1 "the M-learning context" must be explained. Apart from a statement that the data set is large enough, there is no discussion supporting the choice of PLS-SEM as the principal analytic tool. PLS-SEM is well marketed (by marketing academics such as Joseph F. Hair) as a tool for the analysis of causal networks; however, there is little in the way of verification studies to compare how closely the causal relationships identified by PLS-SEM match findings from other methods – typically qualitative.
6. Discussion and conclusion
The first paragraph of this section appears to confuse learning with assessment.
The first paragraph of §6.1 is confused and, in places, garbled. Assertions such as "It is also where the constructivist education theory comes in" and "The practical implications of mobile exam platforms dictate the provision of perfect and safe testing grounds for different types of candidates" reveal a simplistic view of teaching, learning and assessment.
§6.2 does not mention the significant research limitations of the study. The problem with relying on a questionnaire-based research model is that respondents may make statements that are not reflected by their behaviour in practice. Therefore, the findings of this study need to be triangulated against follow-up studies including some or all of the following: (i) observation and monitoring of students' behaviours in the learning phase prior to assessment; (ii) individual interviews and focus groups with students on their own perceptions and feelings; (iii) analyses of students' diaries / learning logs (iv) individual interviews and focus groups with teachers on what they have found to be effective teaching strategies and presentation methods; (v) test results and other records of the educational effectiveness against which the use of smartphone assessment may be compared.
Overall
This paper must be rejected as an educational research study for its reliance on a single data collection source and a single (contested) method of data analysis.
The manuscript has been well checked for use of Academic English and for typographical errors. However, the language register is in places informal and there is a shallowness of critical evaluation. This study appears to have taken a 'systems thinking' view originating from marketing theory and imposed it on an educational context. It reveals a significant lack of understanding of the purpose and process of education.
Author Response

(The authors gave the same response as above.)

Round 2
Reviewer 2 Report
The authors have made a great effort to include guidance for improvement. The manuscript is now more understandable and replicable. There are only a few aspects that should still be considered:
1. Revise the sequential order of citations.
2. Clearly include the collection of the written informed consent of the participants and the approval of the ethics committee.
3. Revise the formatting of some tables with respect to the journal's standards, e.g. Table 7, Tabla 8.
4. Improve the visibility of Figure 1, Figure 2.
5. Review the typeface in new texts as it is not similar to the rest of the manuscript in some words.
6. Include the acknowledgements section.
Once these minor changes have been made, everything would be fine.
Author Response

(The authors gave the same response as above.)

Reviewer 3 Report
In my previous review, I made the point that the interface limitations of smartphones restricts them to text-based assessments such as quizzes, which do not go beyond the Remember and Understand levels of learning (see https://tips.uark.edu/using-blooms-taxonomy/). The Remember and Understand levels predominate in elementary and secondary school education; however, higher education – and especially postgraduate education, which is the context for this study – is more concerned with the Applying, Analysing, Evaluating and Creating levels of learning. Therefore, it should be made clear that assessment with smartphones is of peripheral value in universities and is certainly no substitute for those types of conventional assessment that address higher education. If the paper is to be accepted then these important considerations must be reflected in its Introduction and Discussion sections.
I was very surprised to find that sections of my previous review had been copied and pasted into the revised paper (final paragraph §6.2): this is plagiarism. If the paper is to be accepted then this source must be processed in the same way as any other (i.e. paraphrased or quoted directly with attribution).
Finally, it must be emphasised – in the Abstract, Introduction and Conclusion sections – that this was an initial, exploratory study that needs to be validated by mixed-methods research into not only the effectiveness and validity of smartphone assessment but also its suitability for postgraduate higher education.
Author Response

(The authors gave the same response as above.)
